# Toward equitable coastal community resilience: Incorporating principles of equity and justice in coastal hazard adaptation

Natasha Fox[1] , Jenna H. Tilt[1], Peter Ruggiero[1] , Katie Stanton[2] and John Bolte[1]

[1]College of Earth, Ocean, and Atmospheric Sciences, Oregon State University, Corvallis, OR, USA and [2]Applied Cultural and Environmental Anthropology, Oregon State University, Corvallis, OR, USA

coastal resilience; natural hazards; equity and justice; co-production of knowledge

**Corresponding author:**
Natasha Fox;
Email: foxnat@oregonstate.edu

## Abstract

To meet the challenges of hazards impacting coastal communities, demand is growing for more equitable coastal natural hazard adaptation and disaster mitigation approaches, supported by co-productive research partnerships. This review paper outlines contemporary advances in hazard adaptation and disaster mitigation with attention to how an equity and justice framework can address the uneven impacts of hazards on marginalized and underserved communities. Drawing upon the allied concepts of distributive, procedural, systemic, and recognitional equity and justice, we illustrate how these concepts form the basis for equitable coastal resilience. To demonstrate how equitable resilience can effectively advance contemporary adaptation and mitigation strategies, we present two vignettes where collaborative partnerships underscore how equitable coastal hazard planning and response practices complement these processes in coastal zones subject to large earthquakes and tsunamis. The first vignette focuses on disaster response and takes place in the Tohoku region of Japan, with diverse gender and sexual minority community members' experiences of, and responses to, the 2011 Tohoku disasters. The second vignette centers on hazard planning and takes place on the U.S. Pacific Northwest coast along the Cascadia Subduction Zone to demonstrate how principles of distributive, procedural, systemic, and recognitional equity can inform the co-production of alternative coastal futures that prioritize equitable resilience. From this discussion, we suggest applying an equity lens to research processes, including alternative futures modeling frameworks, to ensure that the benefits of hazard adaptation and disaster mitigation strategies are equitably applied and shared.

## Impact statement

Growing threats posed by natural hazards demand that coastal hazard planning, response, and adaptation practices safeguard coastal communities while minimizing uneven impacts on historically underserved groups. Scientific research partnerships that prioritize sustained, meaningful, multi-stakeholder community engagement through the co-production of knowledge should be considered best practices for policy-relevant research that aims to help communities grow more resilient. The concepts of distributive, procedural, systemic, and recognitional equity and justice illustrate how these partnerships can form the basis for equitable resilience and adaptation. This is especially important in contexts which are vulnerable to both seismic and tsunami hazards, such as the Tohoku region of Japan and the Cascadia Subduction Zone of North America. This review article examines the state of coastal community resilience practices, demonstrating how the concepts of equity, justice, and co-production of knowledge can effectively support coastal hazard resilience practices in hazard-prone communities for equitable futures. The article demonstrates that these principles linked to co-productive research relationships can be powerful tools for achieving equitable coastal community resilience.





## Introduction and background

Human settlements are increasingly located in proximity to coastlines (Oktari et al., 2020), with some 40% of the global population currently living in coastal regions (Reis et al., 2022). Roughly, 90% of cities worldwide are in coastal regions, and some 60% of these are considered at risk of a tsunami (Sundermann et al., 2014; Reis et al., 2022). Coastal communities are also increasingly forced to balance competing demands, such as meeting the housing and public infrastructure needs of growing coastal populations in ways that integrate both scientific knowledge of tsunami hazard zones, and community desires, perceptions, and participation in hazard planning (Herrmann-Lunecke and Villagra, 2020). Many coastal communities face additional barriers to increasing their resilience, including limited financial resources, lack of capacity to interpret and integrate research and data into planning, and challenges in engaging a broad array of community members in these processes (Lipiec et al., 2018).

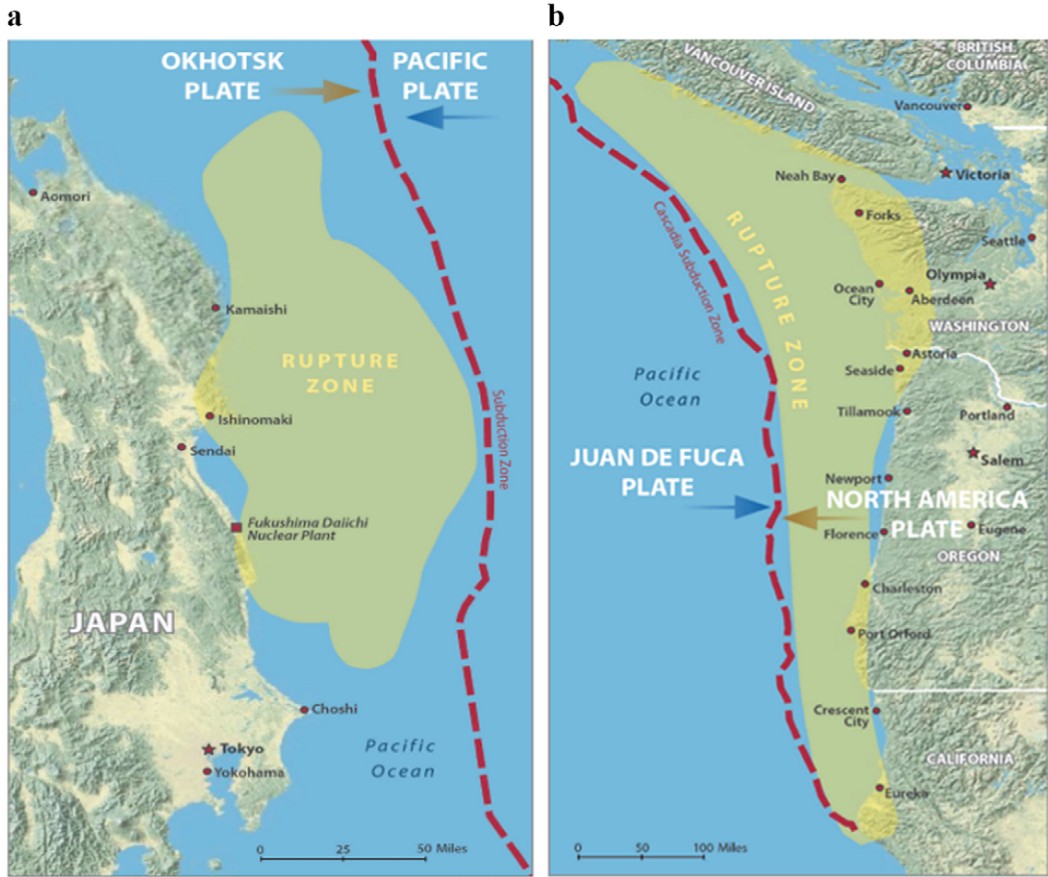

**Figure 1.** (a) Map of the Tohoku region showing the rupture zone in 2011. (b) Map of the Cascadia Subduction Zone (used by permission from the Oregon Department of Geology and Mineral Industries, 2012).

To meet these challenges, demand is growing for the integration of equitable coastal hazard adaptation practices, and scientific knowledge production and policy creation, supported by transdisciplinary co-productive approaches (Peek et al., 2020). This review paper provides an overview of some recent trends in coastal hazard adaptation, a term we use here to encompass the many aspects and temporal contexts of hazard planning, response, and recovery processes focused on achieving coastal resilience. We explore the potential for an equity lens in these processes by putting forth two vignettes that demonstrate how community research partnerships based on equity and inclusion can support equitable coastal resilience. The first example focuses on disaster response and takes place in the Tohoku region of Japan, with diverse LGBT+[1] coastal community members' experiences of, and responses to, the 2011 Tohoku disasters. In the

aftermath of the 2011 Great East Japan Earthquake and Tsunami (GEJE; Figure 1a), LGBT+ communities in the region underwent specific experiences of vulnerability, which led to a set of community responses. These experiences and responses were rooted in everyday marginalization as a stigmatized group, as well as in specific disaster preparations that assumed people's needs would be universal.

The second vignette focuses on coastal hazard planning and takes place on the U.S. Pacific Northwest coast along the Cascadia Subduction Zone (CSZ; Figure 1b) with a geological setting mirroring the Tohoku region. Communities along the Cascadia coastline are highly vulnerable to a disaster similar in impact to the GEJE (OSSPAC, 2021). In the CSZ, experiences from Tohoku can inform geophysical and tsunami modeling and simulations (Frankel et al., 2018; Skarlatoudis et al., 2018), and lead to new approaches to alternative coastal futures modeling. However, coastal hazard policies need to be co-produced with community residents and stakeholders to uphold principles of procedural (Terpstra and Honoree, 2003; Leach et al., 2018), systemic (Bozeman et al., 2022), distributive (Leach et al., 2018; Wiles and Kobayashi, 2020), and recognitional equity (Meerow et al., 2019).

## Contemporary advances in coastal hazard adaptation planning

We begin with a brief overview of recent strategies and scientific advances in coastal hazard adaptation planning. Many of these strategies relate to challenges of urbanization and increasing coastal population density (Sundermann et al., 2014; Reis et al., 2022), with

---

[1]Lesbian, gay, bisexual, transgender, and other gender nonconforming persons. With growing knowledge and awareness of the rich and diverse landscape of genders and sexualities, it is becoming increasingly complex to name all manners in which nonconforming people choose to identify, or not identify as such. In the Japanese language, this terminology is further complicated by the use of katakana alphabet "loan words" such as *erujiibiitii* (LGBT) and other terms. Our use of the term "LGBT+" in this paper follows the use of individuals in both Japan and the USA, who identified with the terms lesbian, gay, bisexual, transgender, or other non-conforming gender identity or sexuality, and who mobilized the LGBT acronym as a strategic political choice when discussing their activities, while acknowledging with the use of the plus ("+") that this acronym is far simpler than the ocean of diversity that exists in the real world of human genders and sexualities.

more recent advances emphasizing the added efficacy of incorporating demographic considerations (Buylova et al., 2020; Maletta and Mendicino, 2022; Reis et al., 2022) and community-collaborative resilience-building (Doyle, 2020; Herrmann-Lunecke and Villagra, 2020; Nakano et al., 2020; Oktari et al., 2020). Advances in scientific and engineering approaches to coastal hazards have better-equipped communities to mitigate the risks associated with the cascading hazards of earthquakes, tsunamis, landslides, and so forth (Satake, 2014; Suppasri et al., 2021; Reis et al., 2022). For example, a growing number of scientific projects, case studies, conferences, publications, field surveys, simulations, and models, many of which proliferated in the wake of recent tsunami disasters, such as the GEJE (Mas et al., 2012; Suppasri et al., 2016; Strusińska-Correia, 2017; Edgington, 2022), have yielded remarkable advances in understandings of tsunamigenic seismic events, as well as early warning systems, building and design codes, and evacuation measures (Makinoshima et al., 2020) to mitigate tsunami impacts around the world (Chock, 2016; Robertson, 2020; Reis et al., 2022). Additionally, the proliferation of knowledge, technology, and planning guidelines around coastal hazards is promulgated at the global scale through institutional frameworks and objectives aimed at anticipating and reducing risk, protecting life and assets, and achieving sustainable long-term socioeconomic development (Shi et al., 2020). Examples include the Sendai Framework for Disaster Risk Reduction (UNDRR, 2015), sustainable development goals (United Nations, 2015), the Paris Climate Agreement (UNFCC, 2015), the New Urban Agenda (United Nations, 2017), and Biodiversity Agenda (Guerquin and Ventocilla, 2020), demonstrating the past three decades' efforts among countries, sectors, and stakeholders to develop new scientific approaches, effective tools and methods, advanced technologies, and streamlined measures that promote coastal hazard adaptation and disaster risk reduction (Shi et al., 2020).

Notable advances have also emerged among federal agencies, organizations, and institutional bodies aimed at reducing the impact of major disasters through both formal and informal interagency collaborations (Bernard et al., 2006; Ward et al., 2018). For example, the Federal Emergency Management Agency (FEMA) recently released guidelines that include a renewed emphasis on community resilience and equity in mitigation planning. This focus is increasingly seen as an important and complementary component, alongside technical advances, in adaptation planning (FEMA, 2022). Recent regional work also acknowledges that achieving tailored multi-hazard solutions that align with the diverse needs of coastal communities requires the co-production of knowledge by a variety of stakeholders, including tribes and governments, professionals, community leaders, and residents (e.g., OSSPAC, 2013; Ruckleshaus Center, 2017; Peek et al., 2020).

## Disaster experiences are not created equal

The growing interdependence of networks and institutions working in natural hazard adaptation planning contexts is accompanied by increasing complexities of societal structures coupled with inherently unpredictable aspects of hazard threats, as a major challenge (Pescaroli and Alexander, 2018; Thiri, 2022). Analysis of hazard risks and adaptation strategies must integrate multiple human, infrastructural, and natural factors that affect the magnitude of risks to be effective. This includes considerations of the reasons why some minority communities opt for alternative sources of post-disaster support (Pescaroli and Alexander, 2018; Kotani et al., 2021; Blagojević et al., 2022; Thiri, 2022), such as out of fear for their personal

safety based on common experiences of marginality in everyday life (Fox, 2020). Thus, new institutional frameworks, analytical tools, and technologies for understanding and mitigating coastal hazard risks have accompanied research findings into the multidimensional societal, and place-based challenges that different groups face during and after disasters (Thiri, 2022).

Despite a growing emphasis on the social dimensions of risk and the inclusion of demographic considerations into disaster planning (Buylova et al., 2020; Maletta and Mendicino, 2022; Reis et al., 2022), the specific incorporation of equity-informed frameworks into these processes has not necessarily kept pace with technological advances (Wisner, 2020). This is particularly concerning as barriers to recovery common among many survivors of disasters (Table 1), such as accessing supplies, information, and resources, are further exacerbated by disability status (Stough et al., 2016; Gaskin et al., 2017; Kosanic et al., 2022), experiencing extreme poverty or being unhoused (Vickery, 2018; Gaillard et al., 2019), non-normative gender identity and gender expression (Fox, 2020; Goldsmith et al., 2021), immigration status (Méndez et al., 2020), age (Malak et al., 2020), and other systemic factors that can conflict, overlap and mutually constitute one another, producing uneven disaster outcomes (Vickery, 2018).

These studies (Table 1) have highlighted the limitations of "one-size-fits-all" approaches to hazard planning that may overlook the diverse needs, capacities, and priorities of marginalized groups (Vickery, 2018; Benevolenza and DeRigne, 2019). However, there are also limits to specificity in coastal hazard adaptation as trying to create policies that can attend to the needs of every individual across a society can seem daunting. Here, the concept of co-benefits, "the secondary or unintended goals of a hazard adaptation project that are additional to the project's primary function, but complementary to its objective of increasing community resilience" (Jones and Doberstein, 2022) can be a powerful tool to maximize resilience across communities. By seeking direct input from communities to identify which needs and potential solutions offer co-benefits beyond a specific marginalized group, policy options become more viable. However, concrete strategies that directly seek out and incorporate knowledge and experiences of underserved communities in hazard planning remain relatively scarce in research and policy (Hiwasaki et al., 2014), and social vulnerabilities like those described above are often difficult to identify and quantify, leading many studies to disregard them altogether (Thiri, 2022). Therefore, a persistent challenge facing coastal hazard planning is how to ensure that technological advances toward hazard resilience are not rendered less effective when communities who are marginalized by oppressive social systems are chronically unable to equally participate in their development and implementation (Kehler and Birchall, 2021). Because vulnerabilities, like disasters, unfold across temporal scales and stages, we draw attention to the importance of an equity lens in all phases of the disaster cycle, including the range of actions taken toward adaptation planning well in advance of a trigger event.

## Key concepts to complement sustainable and resilient coastal hazard adaptation and disaster planning

An equity lens brings many benefits to the nexus of social science, community collaboration, and technical scientific expertise in hazard adaptation planning and decision-making (Brand and Karvonen, 2007; Cote and Nightingale, 2012; Wyborn et al., 2019; Lukasiewicz and Baldwin, 2020; McNamara et al., 2020; Scheidel et al., 2020; Siders, 2022). Hurricane Katrina's

**Table 1.** Examples of studies examining barriers faced by marginalized communities in disasters

| Study | Type | Population(s) of focus | Examples of barriers | Event/hazard type |
|---|---|---|---|---|
| Stough et al., 2016 | Qualitative empirical study | People with disabilities | Lack of access to housing, transportation, employment, physical and mental health, and recovery services | Hurricane |
| Gaskin et al., 2017 | Systematic review | People with disabilities | Lack of access to information, inaccessible evacuation routes, procedures, inaccessible shelters | Climate hazards |
| Kosanic et al., 2022 | Systematic review | People with disabilities | Lack of access to critical information, transport for evacuation and sheltering, lack of accessible beds and bathing facilities | Climate hazards |
| Gaillard et al., 2019 | Qualitative field study | People who are unhoused | Community distrust of official agencies, "rough sleeping" in seismically unsafe structures, stigma, disconnection from media and sources of information | Earthquake/Tsunami |
| Vickery, 2018 | Qualitative study | People who are unhoused | Stigma, loss of campsites, tents, documents, clothing, limited access to shelters | Floods |
| Fox, 2020 | Qualitative field study | People who identify as LGBTQ+ | Lack of access to gender-inclusive bathing and sleeping areas, lack of anti-LGBTQ+ discrimination policies in shelters, difficulty accessing recovery benefits due to exclusionary definitions of "family" | Earthquake/Tsunami |
| Goldsmith et al., 2021 | Review | People who identify as LGBTQ+ | Lack of recognition of LGBTQ+ families in disaster response causes barriers to accessing disaster relief services, prevalence of faith-based organizations in disaster relief, lack of anti-LGBTQ+ discrimination policies in shelters | Climate hazards |
| Malak et al., 2020 | Qualitative field study | Elderly people | Lack of access to specialized medical treatment, difficulties recovering due to fixed income, reliance on family members for information, mobility challenges and physical barriers to shelter access | Cyclones |
| Méndez et al., 2020 | Qualitative empirical study | Undocumented immigrants and people with precarious immigration status | Language barriers to accessing information, ineligibility for federal aid, lack of transportation to evacuate | Fire |

disproportionate impact on Black communities in New Orleans stands out as an example of a missed opportunity for imparting such a lens (Sanchez and Brenman, 2008). While Black communities' reliance on public transit at the time of the 2005 storm was four times higher than that of white communities, rendering many Black residents less mobile, this did not figure into plans for mass evacuation (Pastor et al., 2006). The addition of equity and justice considerations ahead of a hazard's onset, therefore, is of critical importance (Sanchez and Brenman, 2008; Rivera et al., 2022) for equitable resilience.

### Equitable resilience

Resilience has historically been understood as "the ability of a system, community or society exposed to hazards to resist, absorb, accommodate, adapt to, transform and recover from the effects of a hazard in a timely and efficient manner, including through the preservation and restoration of its essential basic structures and functions through risk management" (UNDRR, n.d.). However, this definition can fall short of considering underlying causes of vulnerability, such as those related to equity (Boonstra, 2016; Matin et al., 2018) by either favoring already-advantaged groups and/or prioritizing a return to pre-event unjust social structures resulting in uneven outcomes (Cote and Nightingale, 2012). In some situations, resilience can perpetuate, rather than interrupt, cycles of vulnerability, leaving communities with few tools to transform or adapt to the undesirable circumstances impacting them (Berkes and Ross, 2013; Hardy et al., 2017; Eakin et al., 2021). To operationalize an equitable resilience framework, we first need to understand how different forms of equity can come to bear in different phases of disasters.

Different forms of equity call attention to different aspects of its application, as well as how these principles can operate effectively across different phases of the disaster (preparedness, response, and recovery). Procedural equity refers to how decisions regarding the allocation of risks, resources, and impacts are made (Terpstra and Honoree, 2003; Leach et al., 2018) and focuses on the operation of power. Utilizing principles of procedural equity can highlight how complex administrative systems and structures render specific groups at higher risk than others in a hazard (Rivera et al., 2022), especially during the preparedness phases. A study on procedural inequities illustrates how community development plans placed low-income neighborhoods along the USA–Mexico border at a great disadvantage during recovery from Hurricane Dolly in 2008 (Rivera et al., 2022). Residents who had experienced historic economic segregation and disinvestment, including being routinely denied access to planning and other resources due to the unincorporated status of the neighborhoods, encountered procedural barriers and a lack of clarity around accessing FEMA support after the hurricane. Procedural equity in decision-making around adaptation to hazards requires that communities have the capacity and resources to fully participate in such processes and that non-technical stakeholders' participation is not impeded by the overly complex, highly technical nature of the process (Eakin et al., 2021).

Systemic equity describes the degree to which institutional and administrative resources and policies address the cultural needs of systemically marginalized communities (Bozeman et al., 2022). A study of communities in Baltimore, Kansas City, and Dallas, illustrates the relevance of systemic equity across all phases of disasters. Legacies of racially biased housing and land use practices led to disproportionate heat exposure in poor communities of color (Wilson, 2020). Ongoing historical experiences of systemic inequity

are highly relevant for planners and policymakers in community consultation processes involved in planning and mitigation (Grubert, 2023), and greater consideration of this aspect of equity in hazard response and recovery can ensure that such disparities are addressed rather than deepened (Wilson, 2020).

Distributive equity focuses on the way risks, resources, impacts, and benefits are distributed in society (Leach et al., 2018; Rawls, 2020; Wiles and Kobayashi, 2020). In the hazard context, distributive equity is concerned with the risks and impacts of disasters relative to a community's access to power (Rawls, 2020), with relevance to response and recovery phases, but also during preparation, when the distribution of power, and access to resources across institutional and social networks is key (Doorn, 2017; Eakin et al., 2021). Persistent barriers to low-income and high-minority communities accessing federal assistance after disasters is an example of distributional inequity (Emrich et al., 2022). These can include distributive inequities in disaster recovery assistance that further entrench socioeconomic and racial disparities (Emrich et al., 2022), and systemic inequities in accessing transportation that impede disabled communities from participating in emergency evacuation procedures (Kosanic et al., 2022).

Recognitional equity focuses on how intersecting identities are shaped by historical injustices, which then influence access to resources and differential experiences of vulnerability (Meerow et al., 2019). An Environmental Protection Agency study (EPA, 2006) examining the unintended impacts of redevelopment and revitalization of communities provides an example of recognitional equity. The study demonstrated how projects that disregard local features tied to culture and history, before proceeding with relocation plans and policies, unintentionally contributed to cumulative environmental and health impacts on communities of color (EPA, 2006). Recognitional equity involves careful framing and recognition of place, identity, and social contexts that form the landscape in which people and communities see themselves and interact with policies that impact them (Matin et al., 2018; Wilson, 2020), making this form of equity particularly relevant during the disaster preparedness phase.

Each of these subcategories of equity recognizes the need to intentionally prioritize socially disadvantaged groups by adjusting the rules or structures, or the way resources are distributed, to address the underlying causes of inequality (Wiles and Kobayashi, 2020). Justice can be thought of as the long-term implementation of equitable outcomes by dismantling the societal barriers that cause inequity in all its forms (Lukasiewicz and Baldwin, 2020). As such, justice and equity are interdependent, and linked by geographical and temporal contexts of history and sociopolitical conditions that can hamper or facilitate different communities' access to power (Meléndez, 2020). Issues of justice and equity are always at play in local communities through governance institutions, policymaking bodies, and other systems that allocate resources (Meléndez, 2020), including the consideration and implementation of coastal hazard adaptation planning policies that may make some segments of coastal communities more resilient to hazards than others.

Together, distributional, procedural, systemic, and recognitional equity must form the basis for achieving equitable resilience that, in the long-term, can help to dismantle the social systems that create differentiated outcomes in the first place (Cote and Nightingale, 2012; Pellow, 2017; Davoudi, 2018; Ensor et al., 2021). One point of entry to proceed with the identification and dismantling of social systems that create socially differential outcomes is through equitable co-production of knowledge (Eakin et al., 2021).

## Operationalizing equitable resilience through co-production of knowledge

Involving communities in multi-stakeholder engagement spanning the realms of science, society, and policy is a process known as co-production (Kates et al., 2000; Djenontin and Meadow, 2018; Wyborn et al., 2019). Communities themselves are well equipped to identify and understand the lived experiences and impacts of distributive, procedural, systemic, or recognitional inequities that they face (Goldsmith et al., 2021; Kehler and Birchall, 2021). The development of equitable coastal hazard adaptation planning approaches requires significant community involvement and knowledge to identify and understand potential inequities of existing or proposed approaches, and to co-develop new more equitable mitigation and adaptation approaches and decision-making processes.

Co-production of knowledge operationalizes the fundamental concepts of justice and equity by sharing local knowledge through fair and transparent procedures (e.g., data-sharing agreements, institutional ethics review) and respectfully acknowledging the contributions of marginalized and underrepresented voices by providing equitable compensation to community residents and stakeholders for sharing their knowledge and time through various means (e.g., advisory councils, interviews, workshops). Additionally, co-production entails fundamental changes in decision-making processes (Riccucci and Van Ryzin, 2017), and how science and civil society interact with one another in the world, "through integrating new ways of knowing into new ways of making decisions and acting across all spheres of social, economic, and political life" (Wyborn et al., 2019). For example, Armitage et al. (2011) describe efforts to combine science with local and traditional knowledge to co-manage and co-produce adaptation strategies in Canada's Arctic. The process took place along multiple stages of learning wherein stakeholders were embedded in complex, evolving institutional and knowledge networks. Understandings and framing of problems continuously transformed, eventually co-producing knowledge and institutional arrangements that helped to grow adaptive capacity (Armitage et al., 2011). In this way, processes of co-production can be understood as reflexive, iterative, and dynamic, with diverse forms of knowledge and elements of society continually shaping and reshaping each other (Forsyth, 2004; Linton and Budds, 2014; Wyborn et al., 2019).

## Illustrating an equity and justice framework through two vignettes

The concept of equitable coastal resilience enables a range of frameworks for integrating the goals of marginalized and underserved communities to inform coastal hazards science and potential impacts, and to co-produce adaptation strategies. Below, we provide two vignettes as example applications of these concepts, one from the Tohoku region in Japan post-disaster, and the other from Cascadia (Pacific Northwest Region of the United States) where scientists predict a high likelihood of a catastrophic future earthquake and tsunami. Both Japan and Cascadia sit on similarly active subduction zones (Figure 1), and the information included in the vignettes speaks to different temporal scales of the disasters. The Tohoku vignette illustrates the experiences of a community during and following that event to demonstrate the need for an equity approach to strengthen technological and scientific advances, particularly as they relate to disaster response and recovery. The Cascadia vignette demonstrates how such an approach can be operationalized in the preparation phase.

### The Great East Japan Earthquake

Japan is widely understood to be a world leader in tsunami and earthquake preparation, and a great deal of knowledge and policy-relevant expertise emanates from studies of seismic and tsunami disasters there (e.g., Aitsi-Selmi et al., 2015; World Bank, 2017; Edgington, 2022; Reis et al., 2022). Decades of investment in research, design, education, preparation, and mitigation have led to the world's most sophisticated earthquake and tsunami early warning systems, as well as the strictest seismic building codes, and a nationwide system of drills with high public participation (Bernard and Titov, 2015; Koshimura and Shuto, 2015). These systems are continuously re-evaluated and updated, and each major disaster initiates a new round of research initiatives and overhauls to safety and disaster management (Koshimura and Shuto, 2015; World Bank, 2021). Continuous repeated exposure to disasters has given the population a high level of awareness and knowledge of how to survive a major catastrophe (Aldrich, 2019; World Bank, 2021). However, despite these preparations, Japan was significantly challenged by the low-probability, high-magnitude GEJE disaster (Thiri, 2022). On March 11, 2011, an area approximately 310 miles (500 km) long and 120 miles (200 km) wide slip-ruptured, producing a powerful 9.0 magnitude earthquake that struck the northeastern region of Japan known as Tohoku (Figure 1a). The earthquake caused a highly destructive tsunami which ultimately took the lives of approximately 19,000 people.[2] Sea water flooded roughly 116,000 miles$^2$ (300,000 km$^2$) and created 22 million tons of disaster debris. The tsunami also triggered a nuclear meltdown in Fukushima prefecture, necessitating the evacuation of some 160,000 people in the area and spreading radiation across 5,400 miles$^2$ (14,000 km$^2$). In many locations, initial tsunami wave height predictions were lower compared to the actual heights, owing at least in part to the lack of data available to predict tsunami behavior, including inundation and runup, during such a massive low-probability/high-magnitude event (Mori et al., 2011); many residents erroneously believed that mitigation structures such as concrete seawalls, designed with higher probability, lower impact scenarios in mind, would protect them (Aldrich, 2019).

### LGBT+ community vulnerabilities

At the time of the Tohoku event, marginalized groups in the region, such as elderly people with disabilities, immigrants, and those who identify as LGBT+, faced several vulnerabilities associated with marginalization in the disasters' aftermath (Yamashita, 2012). Members of LGBT+ communities routinely experience discrimination in all geographical locations, including Japan. Different age groups within this community also experience different challenges. For example, LGBT+ seniors are twice as likely to live alone as other seniors (SAGE and National Resource Center on LGBT Aging, 2021) because they are less likely to have children, tend to be more isolated, and lack the support structures that other seniors often benefit from after a disaster. Other subgroups, such as LGBT+ youth, are less likely to have family support and therefore experience higher rates of homelessness than the general population (Keuroghlian et al., 2014). Considerations for the LGBT+ community often go overlooked in disaster plans, even though LGBT+ people make up roughly 9.9% of the population (Dentsu, 2021), and may be particularly vulnerable following a disaster (Yamashita et al., 2017; Goldsmith et al., 2021). Barriers like these call for targeted approaches to public policy because plans for the general

population will not be capable of addressing the specific needs of this community in an emergency.

### Co-production of natural hazard adaptation strategies emerging from LGBT+ experiences of the GEJE

In 2018, coauthor Fox spent 1 year in the Tohoku region conducting original qualitative research to explore the emergence and features of civil society organizations (CSOs) serving local LGBT+ communities post-disaster (Fox, 2020). Relationship-building activities with individuals from local volunteer, civil society, and policy sectors formed the basis for co-productive research relationships. Research questions were co-developed with community participants, and research results were shared with community members for feedback.

This research revealed that across Japan, many LGBT+ people choose to keep their identities hidden in everyday life as a way of avoiding the social stigma still associated with being LGBT+. When the GEJE shook the region, it ruptured the layer of privacy that made this possible. LGBT+ survivors described emergency evacuation centers that lacked privacy barriers and did not have safe and welcoming spaces in which to change clothes, bathe, and sleep (Fox, 2020). Emergency volunteers were untrained and unfamiliar with accommodating LGBT+ survivors. When trying to access local emergency shelters, transgender individuals, for example, reported misunderstandings around whether they belonged in the category of "men" or "women." The policy in many shelters in Tohoku was such that supplies, facilities, and services were divided along binary genders, which alienated people who did not strongly identify with one gender over another. When attempting to locate lost or missing loved ones, LGBT+ survivors were not permitted information about them because they were not considered family by way of Japanese law (Yamashita et al., 2017). These examples illustrate how the lack of an equity lens in coastal hazard planning impacted LGBT+ people's experiences of the disasters, by assuming that needs would be equal across the population.

On one hand, given the strength and impact of the earthquake and tsunami, it is a remarkable testament to advances in adaptation practices that more people did not die as a result. On the other hand, experiences described by members of the LGBT+ community in Tohoku underscore how an arguably "highly resilient" society (with amply funded geotechnical engineering, advanced early warning systems, drills, training, and infrastructure such as seawalls and vertical evacuation structures) still produces uneven outcomes because of a lack of distributional, procedural, systemic, and recognitional equity.

Lessons learned from the 2011 GEJE have helped form the basis for modern tsunami risk management in several ways. Building redundancy (such as backup systems for electricity, water access, communications, and other lifelines) into resilience strategies has emerged as a key lesson from the 2011 disasters (OSSPAC, 2013). The disaster also emphasized the need for both structural mitigations (such as improved construction of seawalls and vertical evacuation towers) and non-structural adaptations (such as improvements to hazard maps, and community education) as a paradigm shift in disaster management (Koshimura and Shuto, 2015). These lessons from the 2011 disasters continue to inform disaster mitigation and adaptation across the world through global institutions such as the Sendai Framework for Disaster Risk Reduction, and partnerships and collaborative international research endeavors (Aitsi-Selmi et al., 2015; Strusińska-Correia, 2017).

Post-2011 lessons relevant to equity in coastal hazard adaptation are also evident. For example, Kumamoto Prefecture's disaster risk reduction plan now makes specific mention of considerations for

---

[2]At the time of writing this paper, the official death toll from Japan's National Police Agency stood at 15,895, with 2,539 people remaining missing.

the local LGBT+ community, noting that "it is necessary to deepen understanding of disaster prevention measures based on the perspective of gender equality, assuming evacuees such as women, children, and sexual minorities from normal times, and to prepare a system for related organizations to cooperate" (Kumamoto Prefecture, 2018). Roughly, 70% of the 47 Japanese prefectures and 47 local governments have updated their regional disaster prevention plans and evacuation center operation manuals to now include considerations for LGBT+ and other sexual minorities in times of disaster (Kyodo News, 2021), and a growing number of emergency evacuation shelters offer access to gender-inclusive facilities. While it is difficult to draw a causal line from post-disaster CSO actions to these targeted policy changes, it is safe to say that LGBT+ CSO post-disaster actions enabled new ways of addressing the underlying web of social and political barriers to LGBT+ resilience in the region.

This vignette illustrates how equitable and just approaches to coastal hazard planning complement and expand important developments in science and technology. The next vignette illustrates the way these forms of equity are being applied toward co-production in the U.S. Pacific Northwest, without the need to experience a disaster first.

### Resilience and vulnerability in CSZ communities ahead of a trigger event

U.S. Pacific Northwest coastal communities also face specific challenges affecting resilience and adaptive capacity. Like Tohoku, the geohazards along the CSZ (Figure 1b) include the threat of a major subduction zone earthquake and a catastrophic tsunami (Oregon Department of Geology and Mineral Industries, 2012). In contrast with communities in Japan's subduction zone (Figure 1a), which have been repeatedly exposed to earthquakes and tsunamis over time, knowledge and experience of impacts of a CSZ event (megaquake and associated tsunami) are limited to Indigenous oral traditions (Losey, 2022), and what scientists have learned in recent decades (Goldfinger et al., 2012). Local livelihoods connected to coastal land and water developed without regular exposure to and experience of earthquakes and tsunamis as fundamental parts of life in this coastal region. Thus, the task of planning for these hazards and adapting CSZ coastal communities to be more resilient is daunting.

However, CSZ communities with partnerships from state and federal agencies have made much progress toward CSZ mitigation, particularly in the realm of public education/awareness and evacuation procedures such as the Tsunami Safe Haven Hill mitigation project in Newport, Oregon (FEMA, 2021), and the "Beat the Wave" evacuation route maps produced by State of Oregon Department of Geology and Mineral Industries (DOGAMI) that show the quickest routes out of the tsunami inundation areas (Priest et al., 2016). Additionally, large-scale hazard-resilient infrastructure investments have been made with the building of the first vertical evacuation structures in North America (Ocosta School in Westport, Washington, the Gladys Valley Marine Studies Building in Oregon, and the Shoalwater Bay Indian Tribe in Washington). Moreover, the town of Seaside Oregon recently relocated their K-12 public schools outside of the tsunami inundation zone. Statewide regulatory policies have been enacted with the passing of Oregon Senate Bill 379 in 1995 which required new critical facilities (e.g., schools, fire and police stations, hospitals) to be built outside the tsunami inundation zones (Oregon State Legislature Archives, 1995). However, Senate Bill 379 was repealed in 2019, leaving few statewide legislative mechanisms in place for CSZ hazard mitigation and adaptation for Oregon's coastal communities (OSSPAC,

2021). Future adaptation policies will require sustained engagement with a broad array of stakeholders, including tribes and other governments, professionals, community leaders, and coastal residents to ensure equitable distribution of adaptation practices and their intended impacts.

Key economic drivers in CSZ coastal communities include seaports and fisheries industries, timber mills, and more recently, tourism (Lewis et al., 2019). The labor force for these industries is primarily low-wage and comprises a large percentage of Latinx coastal residents. Approximately, 32% of employees in agriculture, forestry, fishing, and hunting industries and 18% of employees in accommodation and food service industries identify as Hispanic or Latinx (Procino, 2022). Due to the specific needs and requirements of these sectors, these workplaces are in high-hazard risk areas along coastal shores and the bay fronts most vulnerable to tsunami inundation (CAUSA, 2012). Understanding Latinx needs and perceptions regarding hazard preparedness and response is a fundamental first step in identifying equitable and just adaptive strategies for CSZ coastal communities.

Working with trusted community partners, we held a series of discussions with Latinx coastal community members, the majority of whom worked in fisheries and service sector industries highly exposed to natural hazards (Stanton and Tilt, 2023). Through these discussions, we learned that Latinx residents would turn to trusted community organizations such as churches, non-profits, and community centers during times of need, including the aftermath of a CSZ event (Stanton and Tilt, 2023). These community assets were more associated with having resources to help them and were seen as more welcoming spaces than locations typically associated with emergency response such as fire and police stations, and hospitals —the very places that were under the purview of former Senate Bill 379 for adaptive protective measures. While critical facilities are essential to disaster response, recognizing that some of the most marginalized and underrepresented populations are hesitant to utilize these facilities demonstrates the need for broader discussion regarding how existing mitigation and adaptation strategies could be made more robust by implicitly incorporating principles of equity.

However, this knowledge does not help decision-makers choose what critical facilities or community assets to protect through adaptation measures, such as relocation of a critical facility or community asset out of the inundation zone, especially with limited resources at the local and state level. How can these decisions be made in a just and equitable way? This "decision-makers dilemma" is not unique to Cascadia coastal hazards but is a dilemma facing local decision-makers everywhere making adaptation choices (Siders, 2022). Navigating this dilemma could be made less daunting by utilizing a policy framework that centers on equity, such as Targeted Universalism (Powell, 2008; Powell et al., 2019). In the following section, we provide a conceptual framework that incorporates some of the principles of Targeted Universalism to guide equitable and resilient coastal futures through adaptation decision-making support. While decision-making support tools are numerous, here, we focus on agent-based modeling (ABM) to illustrate the conceptual framework because ABM allows modelers to set agent rules of behavior that can be guided by equity principles.

### Building a conceptual framework for equitable and resilient coastal futures

Targeted Universalism, sometimes referred to as "Equity 2.0," (Powell, 2008; Powell et al., 2019) stems from public health policy

that combines "universalism"—policies that treat all individuals equally, regardless of race, class, and sexual orientation, and so forth, such as minimum wage, universal health care (Bagenstos, 2014), with targeted policies that provide protections or benefits to a specific population segment. Examples of such targeted policies include programs like Supplemental Nutritional Assistance Program (SNAP) and the Americans with Disabilities Act in the United States. On their own, both policy approaches can be problematic: universal policies do not guarantee that the fundamental policy goals will be met. For example, providing universal health insurance does little good if no health care facilities are nearby or provide bilingual services (Milstein et al., 2010). Targeted policies are often perceived as unfair because they do not apply to everyone and are vulnerable to resource reductions or repeals (Grier and Schaller, 2020).

The concept of Targeted Universalism was developed to address these problems by applying a universal goal that is achieved through multiple tailored policies that consider systemic and situational circumstances or structures that limit progress toward the shared, universal goal. This shared universal goal is not set based on what the advantaged groups already have (e.g., "closing the achievement gap"), but rather on what is desired by society, such as a higher standard a performance (e.g., education, quality of life) for everyone, regardless of background. A case for Targeted Universalism has been made for achieving educational and behavioral standards for youth (Farmer et al., 2022), as well as applications of reaching COVID-19 goals (Gaynor and Wilson, 2020). Other similar policy frameworks, such as Proportionate Universalism (Carey et al., 2015), also strive to balance universal and targeted policies but do not emphasize the universal goal. In addition, this framework relies on local governance for implementation that may have limited capacity or may not recognize the systemic issues limiting the distribution of goods and resources. Therefore, we advocate for the Targeted Universalism approach because it allows for incorporating a diversity of policy options tailored to address specific group needs that cumulatively add to the progression of the entire community, region, or state toward the universal goal of greater coastal resilience.

Understanding and analyzing the potential impacts of employing a Targeted Universalism approach to future coastal resilience is facilitated by utilizing a variety of decision-support tools such optimization, cost–benefit analysis, multi-criteria decision analysis, structured decision-making, adaptive management, and scenario planning. In particular, ABM is a useful approach to visualize the impacts of different coastal hazard adaptation strategies and has been used to understand natural hazard adaptation to flood risks, droughts, and other hazards (see Zhuo and Han, 2020; Schrieks et al., 2021; Di Noia, 2022 for reviews of ABM on these topics). Within Cascadia, ABM has been used to model alternative future scenarios with varying coastal chronic hazard adaptation strategies (Lipiec et al., 2018; Mills et al., 2018, 2021) and for tsunami evacuation scenarios (Mostafizi et al., 2017; Wang and Jia, 2022). Yet, most ABM model developers do not incorporate concepts of procedural, systemic, distributive, and recognitional equity in model development (Voinov et al., 2016). ABM, however, has the potential to incorporate the principles of Targeted Universalism because of its flexible approach to the assignment of individual characteristics to specific agents and inherent focus on distributional outcomes (Williams, 2022).

The conceptual framework for equitable and resilient coastal futures presented below (Figure 2) asks fundamental questions for researchers to consider regarding distributive, procedural, systemic, and recognitional equity during each phase of an alternative futures modeling process based upon the key principles of Targeted

Universalism (Powell, 2008; Powell et al., 2019). Each of these phases provides ample opportunities for the co-production of knowledge between disciplines (e.g., natural sciences, social sciences, engineering), community members, and stakeholders. Below, we provide an in-depth description of the conceptual framework (Figure 2).

A key step of Targeted Universalism is to identify and understand specific groups that may run counter to dominant norms or policies (Figure 2, Box A). From the equity literature synthesized above (e.g., Terpstra and Honoree, 2003; Leach et al., 2018; Meerow et al., 2019; Wiles and Kobayashi, 2020) we suggest key questions to drive data collection for alternative coastal futures modeling including: *How are vulnerabilities and adaptations distributed throughout the community* (distributional equity)? *What adaptation strategies would reduce vulnerabilities and why* (distributional equity)? *What drives/perpetuates these vulnerabilities* (systemic equity)? *How have adaptation decisions been made in the past* (procedural equity)? *Who/what values are being prioritized in adaptation decisions* (recognitional equity)? These equity questions tailor the broad Targeted Universalism assessment to specific coastal community resilience issues. For example, in the CSZ vignette, Latinx communities exhibited hesitancy in utilizing critical facilities and stated their preference to rely on community assets in times of disasters (Stanton and Tilt, 2023). Similarly, in the Tohoku vignette, LGBT+ survivors of the 2011 earthquake and tsunami were hesitant to access existing shelters and emergency services due to fears for their safety and privacy (Fox, 2020). Both examples point to the importance of understanding the diverse needs and values of marginalized and underrepresented community members and incorporating these needs and values into adaptation planning.

Williams (2022) provides examples of ABM that have incorporated recognitional, procedural, and distributional equity. However, the review does not include aspects of systemic equity that are foundational to other equity lenses (Wiles and Kobayashi, 2020). For example, the most abundant use of equity in ABM is to stratify ABM outcomes based on socio-demographic variables (e.g., distributional equity; Williams, 2022); yet, what is not addressed is the systemic underpinnings of uneven distribution of resources. To address this gap, we have developed a set of equity-driven questions to consider during alternative futures model development (Figure 2, Box B) that expands upon Williams , 2022. These questions include: *How does adaptation to one risk relate to other natural hazard risks (e.g., maladaptation or co-benefits)* (distributional equity)? *What landscape processes need to be modeled to capture these multi-hazard risks* (distributional equity)? *What underlying land use characteristics and/or data gaps drive/perpetuate vulnerabilities* (systemic equity)? *What data proxies are available and acceptable to use* (systemic equity)? *What adaptation scenarios are modeled and why* (procedural equity)? *How does modeler positionality factor into the modeling* (procedural equity)? And *who's/what values are being prioritized in adaptation scenarios* (recognitional equity)?

Application of these equity-driven questions is critical when developing adaptation scenario models that are often driven by data availability that may mask marginalized or underrepresented groups. For example, identifying LGBT+ households from the U.S. Census can be problematic due to how definitions of "households" and "families" have changed over time (Deng and Watson, 2023; U.S. Census, 2022) and in Japan, robust and generalizable household and population data on sexual orientation and gender identity is not routinely collected.

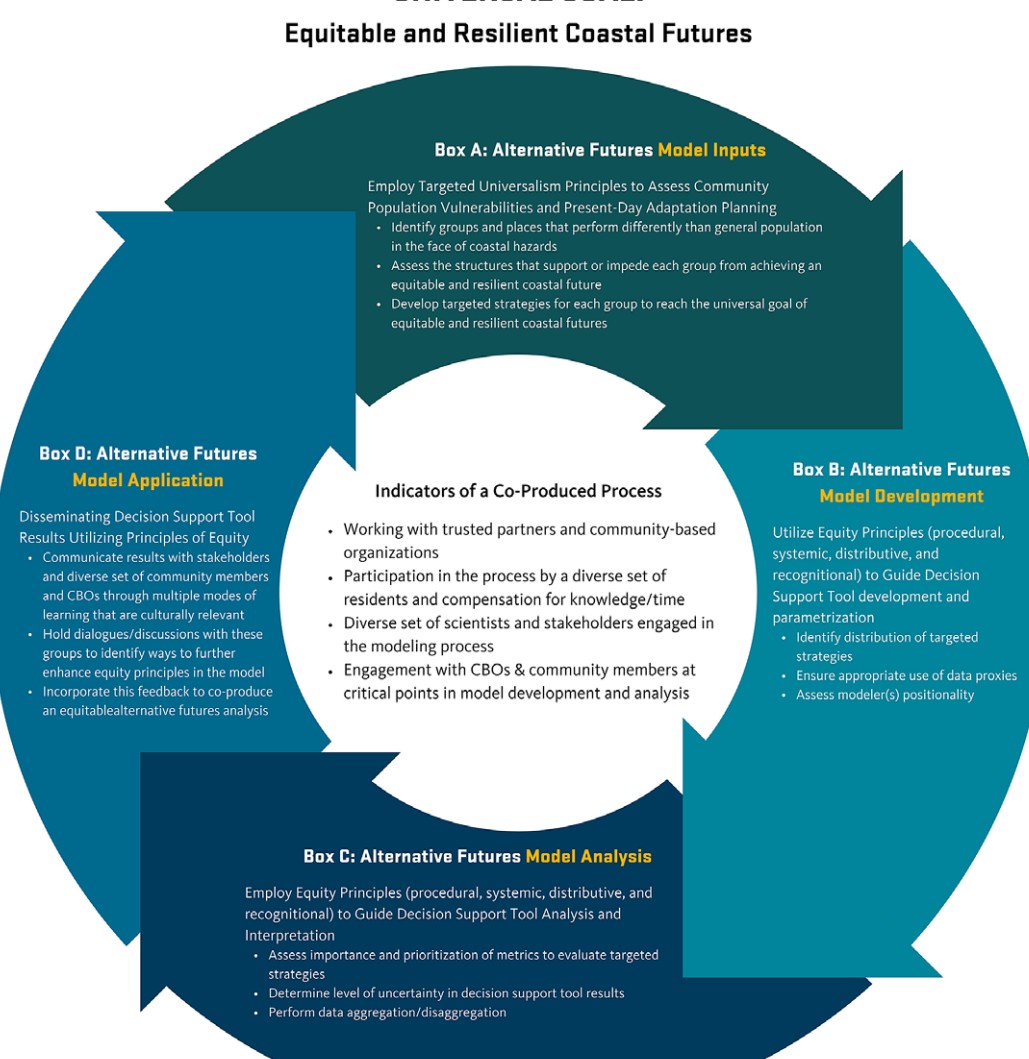

**Figure 2.** A conceptual framework for equitable and resilient coastal futures. The proposed conceptual framework applies Targeted Universalism for policy development (Powell et al., 2019) to an agent-based modeling approach (e.g., alternative futures modeling) to develop targeted coastal hazard adaptation strategies that account for a diverse set of marginalized and underrepresent population needs. The conceptual framework is guided by co-production of knowledge to ensure diverse community voices and values are in embedded in each phase of the process.

Furthermore, many ABMs incorporate optimization of a specific goal or metric, such as cost, life-safety, and so forth in their analysis of alternative adaptation strategies or futures (Figure 2, Box C), (for review, see Barbati et al., 2012). This optimization analysis should also include distributional equity questions such as: *What metrics of impacts are important to communities and why?* And *how are adaptation scenario impacts distributed across the community?* Modelers may choose not to include some variables due to large amounts of missing data or they may decide to aggregate other datasets together. However, these decisions could mask key equity concerns related to who is benefiting (or not benefiting) from specific adaptation scenarios (Obermeyer et al., 2019). Therefore, researchers should ask: *How do model assumptions drive impact performance* (systemic equity)? *What level of uncertainty is acceptable to the community* (systemic equity)? *What impacts are analyzed and why* (procedural equity)? And *what impacts should be prioritized (recognitional equity)?*

Dissemination of results can foster systemic inequities regarding science education and understanding (Polk and Diver, 2020), and therefore requires an inclusive science communication approach (Márquez and Porras, 2020) (Figure 2, Box D); particularly regarding *how* scenarios are presented, especially when communicating risk probabilities and uncertainties of ABM results. Such an approach centers the person and community first and foremost (Polk and Diver, 2020), provides information in multiple languages and is culturally relevant (Márquez and Porras, 2020), and offers multiple modes of learning and engagement, such as gaming (Hobbs et al., 2019).

The following questions can help guide researchers through the dissemination of model results (Figure 2, Box D): *How are model results distributed/communicated throughout the community* (distributional equity)? *What drives understanding of the results and perception of risks/probabilities* (systemic equity)? Additionally,

what feedback is incorporated in refining the model is a key aspect of procedural equity. Researchers should consider *whose values are being prioritized in communicating model results and incorporating feedback into the model* (recognitional equity). And *who gets to change the model and why* (procedural equity)? When incorporating feedback from stakeholders and community members. Most importantly, the co-production of knowledge and understanding of ABM results should not favor one group over another. Discussions with decision-makers must include a comprehensive understanding and assessment of who, and who is not, represented in that decision-making space (Meléndez, 2020).

Co-production of knowledge is integral to the development, recruitment, and analysis of community values, needs, and perceptions that inform alternative futures model development in all phases (Figure 2). Voinov et al. (2016) reviewed modeling approaches used in ABM and found that indicators of a co-developed model process are most apparent in the early stages of the model development (e.g., scoping and data collection) and evaluation of the model outputs of project outcomes; thus skipping key steps in model development, refinement, and analysis (Voinov et al., 2016). Yet other studies find that when community members and stakeholders are fully involved in co-production in all stages of the modeling process, the models can act as "boundary objects" (e.g., objects that bridge divides between groups of people and the values/perceptions they hold) and facilitate new knowledge generation (Voinov et al., 2016; Lemos et al., 2018; Tilt et al., 2022). In the conceptual framework presented in Figure 2, the co-production of the knowledge process can be expanded by engaging community members and stakeholders to modify model parameterization and development to assess and evaluate use of proxies, metrics, and the impacts of data aggregation/disaggregation, and uncertainties.

While the co-production of knowledge is iterative, place or interest-based, and no single recipe exists for successful co-production (Cooke et al., 2021), Mach et al. (2020) provides key goalposts for evaluating actionable-knowledge production: 1) substantive interactions between all involved; 2) ensuring equitable relationships between parties in the process; and 3) producing knowledge that is usable by decision-makers (Mach et al., 2020). From these studies, and many others (see Will et al., 2020; Steger et al., 2021; Weiskopf et al., 2022), we see that key indicators of co-produced ABM processes include: working with trusted partners, such as community-based organizations; compensation to community members; strong recruitment of diverse community members; and opportunities to check, respond to, and validate the data gathered, as demonstrated above in the Tohoku and CSZ vignettes.

In summary, the conceptual framework presented in Figure 2 requires iterative stages of co-production of knowledge with community members, stakeholders, and others to develop equitable decision support procedures that can identify a range of community needs and values and apply these needs and values to specific adaptation strategies that will make progress toward the universal goal of an equitable and resilient coastal future. The vignettes from Cascadia and Japan provide an example of co-productive knowledge inputs to guide the development of alternative coastal futures modeling using the conceptual framework.

## Discussion and conclusion

To grow equitable coastal community resilience to a range of hazards, universal hazard adaptation and risk mitigation

technologies and expertise can be made more effective by employing targeted strategies that address underlying causes of vulnerability (Matin et al., 2018; Meerow et al., 2019). As vignettes focused on the LGBT+ community in Tohoku, Japan and the Latinx community in the CSZ illustrate, creating a plausible future adaptation that encompasses principles of equity requires sustained co-productive engagement with community members and an in-depth understanding of the social processes that underpin vulnerability across different demographics–including those that relate to coloniality, racism, homophobia, and others. The co-productive process in both cases generated dynamic cross-sectoral interactions, rich qualitative data, and a set of findings and recommendations co-developed by and for members of the community (Fox, 2020; Stanton and Tilt, 2023).

Intentionally including members of underrepresented communities in multiple steps in the research and planning process can be an effective means of growing equitable resilience. The conceptual framework provided here (Figure 2) offers an example of how iterative stages of co-production of knowledge with community members and other stakeholders can be used to develop more equitable alternative coastal futures modeling procedures by representing a range of coastal futures scenarios, as well as mechanisms to evaluate the impact of those scenarios. Key stages of model development from data inputs, analysis, and dissemination of results are critical points to incorporate the co-production of knowledge to achieve equitable outcomes. Thus, equitable coastal resilience can be thought of as both a process and an outcome where scholars, decision-makers, and diverse community members collectively (and iteratively) ask "resilience of what, to what, and for whom?" (Cretney, 2014; Meerow and Newell, 2019). These fundamental questions should drive coastal hazard planning, research, and adaptation practices if we are to meet the growing challenges presented by coastal hazards in equitable ways. In doing so, scientists and practitioners can better align the goals of coastal hazard planning and adaptation with the everyday needs and contributions of marginalized and underrepresented groups to achieve more just, equitable, and resilient coastal futures.

**Open peer review.** To view the open peer review materials for this article, please visit http://doi.org/10.1017/cft.2023.24.

**Acknowledgments.** We acknowledge funding that supported this research in part from Oregon Sea Grant under Award NA18OAR170072 (CDFA 11.417) from the National Oceanic and Atmospheric Administration's National Sea Grant College Program and the Cascadia Coastlines and Peoples Hazards Research Hub, an NSF Coastlines and People Large-Scale Hub (NSF #2103713). N.F. also undertook Japan-based fieldwork supported by funding from the 2017 Japan Foundation Doctoral Research Fellowship.

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
