## [Reviewer Report]

Manuscript discusses disaster risk reduction and preparedness for transforming natural hazards influenced by climate change. The concept of equitable resilience in coastal development is then introduced using the Japanese Tohoku region and US Cascadia Subduction Zone examples to highlight the challenges faced by LGBT+ and Latinx communities, respectively.

The work proposed for publication meets UNESCO-UNDRR global agendas towards disaster risk reduction, while focusing on an emerging hot-topic: equity in risk management policies. However:

• the review of the state of the art is insufficient to constitute a revision article,

• and it is followed by a generic conceptual framework that does not provide sufficient detail or novelty to qualify as a research paper.

Overall, the manuscript tackles an important topic but is too low level to be of practical use and does not provide any real help or guidance for risk management. Hence, I recommend major revisions and one of the two hypotheses: 1) enhance the revision nature of the manuscript and submit as a revision paper (if the journal accepts such type of publication), or 2) improve the description of the methodology used to tackle the CSZ vignette, its numerical application to the case-study, data used and, quantification and discussion of the results.

General comments

G1 – The manuscript fits within the scope of the journal and raises awareness of the importance of equity in disaster risk reduction policies.

G2 – The abstract and title are reasonably aligned with the content of the manuscript. Both refer to future equitable principles within the framework of coastal risk management. However, after reading the manuscript, both the title and abstract seem overstated. This can lead to high expectations that the manuscript cannot fulfill.

G3 – The literature review would benefit from complementary references on two main aspects.

Prior to identifying the limitations of current risk management policies, it would be useful to start with a brief outline of the most common mitigation strategies. Typically, these strategies are related to urbanism and structural performance criteria, but more recent advances now take into account demographic factors.

In addition to the above, I would also like to see a reference to other minority and under-represented groups in the manuscript (even if only citing literature and mentioning their importance for future integrative research in the framework of risk policies).

G4 - The conceptual methodology is (briefly and) poorly described and ignores interactions that have the potential to greatly influence the efficiency of risk management policies, such as cumulative effects associated with multi-hazard dependencies.

G5 – Figures and captions need careful revision and/or replacement to provide clear and straightforward information to the reader. Since Figures 1 and 2 are taken from other works, the quality of the image is, from my point of view, unacceptable for publication purposes. Since none of them are original, consider replacing them with more illustrative and contemporary works. Even Figure 3, the only original figure in the manuscript, has very poor resolution (and personally, I cannot understand what it aims to represent). On the other hand, the authors collected data from community members that could be used to generate intuitive graphic information.

G6 – In resume, the manuscript needs major revisions before it is publishable.

Major comments

There are 6 major issues with the manuscript that must be addressed.

Major Q1 – The Introduction and Background section starts with global warming and climate change and quickly moves to earthquake and tsunami threats.

What is the link between long-term climate change and abrupt earthquake and tsunami hazards? The direct connection between climate changes and earthquakes is weak, and most likely to have some influence on micro-seismicity. For tsunamis, the rising of average sea level due to climate change (order of centimeters) is nearly negligible when compared with tidal variations (order of meters from low to high sea levels).

The manuscript needs a discussion in order to understand and accommodate the influences of climate change on coastal hazard, risk and resilience contexts. In addition, how does it extrapolate to the two vignettes used to discuss coastal risk mitigation policies?

Major Q2 - The terminology (Key concepts) is compiled from definitions and published work. I can understand it as part of the contextualization. Additionally, it is of the utmost importance to identify gaps in risk strategies towards non-discriminatory policies, such as the ones highlighted in the manuscript. However, I cannot accept that remarkable advances in coastal risk management, in particular for earthquake and tsunami multi-risk, are here completely ignored before being criticized for their shortcomings. Many scientists and engineers dedicate a career lifetime to understanding and characterizing the (very complex) physical phenomena associated with hazard generation, propagation, and interaction with natural and built environments before it is a product to feed the fields of social sciences, decision making, and politics. These efforts should be acknowledged in the manuscript as the cornerstone of any further development.

Multi-risk results from a geophysical understanding of natural hazards, field experts surveying the field after a catastrophic event, large-scale lab campaigns and sophisticated numerical models to better understand physical processes, and characterization of natural and built environments' responses to extreme multi-hazard events, as well as the definition of criteria for their performance. It encompasses a universe of multidisciplinary and transversal collaborations that have already overcome many challenges towards the mitigation of cascading earthquake and tsunami effects on coastal communities.

I suggest the authors revise the trend in global planning. It is the Sendai Framework for Disaster Risk Reduction that constitutes the roadmap, but its connection to other global agendas is key. Examples of international agencies and instruments include the Sustainable Development Goals, the Paris Climate Agreement, the New Urban Agenda and the Biodiversity Agenda.

I suggest the authors revise projects funded to develop multi-risk management, with a particular focus on strategies for megathrust scenarios. The significant outcomes of the work developed by national and international experts and committees are already helping coastal communities mitigate the consequences of cascading earthquakes and tsunamis. Many strategies have been implemented in the form of early warning systems (currently covering all oceans) and governmental regulations for evacuation procedures and ensuring structural performance criteria. Examples:

• World Association for Waterborne Transport Infrastructure

• US guidelines from the American Society of Civil Engineers, and the Federal Emergency Management Agency,

• Japanese Ministry of Land, Infrastructure, Transport and Tourism guidelines

• The newly formed European FIB Task Group 2.13, etc.

I suggest the authors revise literature that compiles the state of the art regarding cascading earthquake and tsunami multi-risk. Some examples (by newest):

• Reis, C., Lopes, M., Baptista, M. A., & Clain, S. (2022). Towards an integrated framework for the risk assessment of coastal structures exposed to earthquake and tsunami hazards. Resilient Cities and Structures, 1(2), 57-75. doi:10.1016/j.rcns.2022.07.001

• Oktari RS, Syamsidik, Idroes R, Sofyan H, Munadi K. City resilience towards coastal hazards: an integrated bottom-up and top-down assessment. Water 2020;12(10). doi:10.3390/w12102823.

• Buylova A, Chen C, Cramer LA, Wang H, Cox DT. Household risk perceptions and evacuation intentions in earthquake and tsunami in a Cascadia Subduction Zone. Int J Disaster Risk Reduct 2020;44:101442. doi:10.1016/j.ijdrr.2019.101442.

• Wisner, B. (2020). Five years beyond Sendai—Can we get beyond frameworks?. International Journal of Disaster Risk Science, 11, 239-249. doi:10.1007/s13753-020-00263-0

• Maletta R, Mendicino G. A methodological approach to assess the territorial vulnerability in terms of people and road characteristics. Georisk 2020;0(0):1–14. doi:10.1080/17499518.2020.1815214.

• Doyle, E. E., Lambie, E., Orchiston, C., Becker, J. S., McLaren, L., Johnston, D., & Leonard, G. (2020). Citizen science as a catalyst for community resilience building: A two-phase tsunami case study. Australasian Journal of Disaster and Trauma Studies, 24(1), 23-49.

• Shi, P., Ye, T., Wang, Y., Zhou, T., Xu, W., Du, J., ... & Okada, N. (2020). Disaster risk science: A geographical perspective and a research framework. International Journal of Disaster Risk Science, 11, 426-440. doi:10.1007/s13753-020-00296-5.

• Herrmann‐Lunecke, M. G., & Villagra, P. (2020). Community resilience and urban planning in tsunami‐prone settlements in Chile. Disasters, 44(1), 103-124. doi:10.1111/disa.12369.

• Pescaroli, G., & Alexander, D. (2018). Understanding compound, interconnected, interacting, and cascading risks: a holistic framework. Risk analysis, 38(11), 2245-2257. doi:10.1111/risa.13128.

• Poljansek, K., Marín Ferrer, M., De Groeve, T., & Clark, I. (2017). Science for disaster risk management 2017: knowing better and losing less. ETH Zurich. doi:102788/688605. ISBN 9789279606786

• Satake, K. (2014). Advances in earthquake and tsunami sciences and disaster risk reduction since the 2004 Indian ocean tsunami. Geoscience Letters, 1(1), 1-13. doi:10.1186/s40562-014-0015-7.

• Bernard, E. N., Mofjeld, H. O., Titov, V., Synolakis, C. E., & González, F. I. (2006). Tsunami: scientific frontiers, mitigation, forecasting and policy implications. Philosophical Transactions of the Royal Society A: Mathematical, Physical and Engineering Sciences, 364(1845), 1989-2007. doi:10.1098/rsta.2006.1809.

Major Q3 – Tohoku and Cascadia subduction regions. The novelty, applicability and scalability of the work that is presented in the submitted version of the manuscript is not enough to be publishable. And that would represent a lost opportunity to raise awareness of the need to address inclusive strategies in risk management policies. Therefore, one of the ways to add value to the manuscript is to actually present a comprehensive review of the literature.

Q3.1 – Besides the ones previously suggested in Q.2., the review should then address complementary insights into the challenges minority groups face. It is, after all, the manuscript’s objective key. For the sake of coherency on DEI principles, at least refer to other groups for which natural risk poses additional barriers to transpose (limited mobility, poverty, illiteracy, …). One possible example:

• Stough LM, Kang D. The Sendai framework for disaster risk reduction and persons with disabilities. Int J Disaster Risk Sci 2015;6(2):140–9. doi:10.1007/s13753-015-0051-8.

Q3.2 – Japan’s preparedness for megaquakes. I suggest a short text discussing:

• Why is Japan considered the best-prepared nation in the world?

• How preparedness was jeopardized when the hazard estimates were exceeded in 2011?

• What were the main lessons learned from 2011? and

• How did these lessons become the basis of modern tsunami risk management?

By understanding and learning from the Japanese event, one can assign a more fair perception to sentences such as ‘demonstrate how a highly resilient society can transcend past tragedies...’ and ‘...without the need to experience a disaster first’.

Q3.3 – Cascadia’s description. ‘…The science and our knowledge of expected impacts of a CSZ event (megaquake and associated tsunami) are still relatively new (Goldfinger et al., 2012).’ The amount of programs, such as NHERI and Cascadia CopeHub, and the number of publications and conferences on the topic show otherwise.

‘…allows for broader discussion regarding who is benefiting from proposed mitigation and adaptation strategies…’ – apparently everyone does except ‘…some of the most marginalized and underrepresented populations…’. Consider adding a brief discussion on the reasons influencing people to choose alternative post-disaster support, analyzing the trade-off between enhancing the (inclusive) education of populations or assure that community centers are prepared to play the role of shelter. Some references (and references therein):

• Thiri, M. A. (2022). Uprooted by tsunami: a social vulnerability framework on long-term reconstruction after the Great East Japan earthquake. International Journal of Disaster Risk Reduction, 69, 102725.

• Wood, N., Jones, J. M., Yamazaki, Y., Cheung, K. F., Brown, J., Jones, J. L., & Abdollahian, N. (2019). Population vulnerability to tsunami hazards informed by previous and projected disasters: a case study of American Samoa. Natural Hazards, 95, 505-528.

• Kotani, H., Tamura, M., Li, J., & Yamaji, E. (2021). Potential of mosques to serve as evacuation shelters for foreign Muslims during disasters: a case study in Gunma, Japan. Natural hazards, 109(2), 1407-1423.

• Blagojević, N., Didier, M., & Stojadinović, B. (2022). Quantifying component importance for disaster resilience of communities with interdependent civil infrastructure systems. Reliability Engineering & System Safety, 228, 108747.

• https://reliefweb.int/report/world/tsunami-hate-and-xenophobia-targeting-minorities-must-be-tackled-says-un-expert

• https://news.un.org/en/story/2021/03/1087412

• https://www.undrr.org/publication/marginalized-and-minority-groups-consideration-ndra

‘To progress towards a more equitable coastal community resilience requires co-development of such strategies, as well as specific metrics used to gauge the potential impacts of those strategies prior to implementation.’ This sentence also allows us to discuss the importance and efficiency of having a mitigation plan versus having none. Again, please recognize the importance of the whole risk management process before highlighting possible points of improvement.

Major Q4 – The expansion of the conceptual framework. ‘To answer this call, we have expanded upon ‘envisioning coastal futures’ conceptual framework (Bolte et al., 2007; Lipiec et al., 2018; Mills et al., 2018; Mills et al., 2021) that models landscape processes, to include both chronic (e.g., sea-level rise) and acute (e.g., a CSZ, magnitude 9 event), with socioeconomic information to explore future conditions based on a series of hazard and policy scenarios (Figure 3).’.

Q4.1 – The methodology needs additional explanations of its constitutive processes. Is it a probabilistic process? What’s behind Envision’s quantitative solutions? And, if Envision was used to model the case-study, why are there no quantitative results in the present manuscript?

Q4.2 – Fig. 3. is overall very confusing. What’s the link between chronic and acute scenarios? What’s a CSZ scenario? The CSZ fault system is always there: usually with less tectonic activity and rarely with extremely active behavior, as experienced in the past.

Q4.3 – ‘In this case, we have co-developed targeted strategies that focus on building adaptive capacity for critical assets deemed inclusive to Latinx coastal community members – either through relocation of those assets to safer areas outside the inundation zone (Realign scenario) or fortifying protection of those assets through building retrofits (Protect scenario).’ A local/regional tsunami (as inherently associated with CSZ), causes two effects on coastal structures: strong ground motions and tsunami effects. From a structural performance perspective on buildings serving as shelters, it is necessary to account for two possibilities. One is for structures located outside the tsunami-inundated area. In this case, buildings have to be designed to support a high magnitude earthquake (energetic enough to trigger a tsunami!) so it is safe to serve as shelter for people. The second is for structures located in an area prone to tsunami inundation. Shelter for people can be provided by vertical evacuation buildings designed to withstand cascading ground movements and tsunamis. But other structures, even if retrofitted, cannot guarantee the criteria for immediate occupancy. How exactly do the ‘realign’ and ‘protect’ scenarios work? Are these scenarios inextricable, complementary or individual?

Q4.4 – Later in the manuscript, it states ‘…we have not only identified specific community assets utilized by marginalized populations, and their spatial locations…but have also collected information regarding the structural quality of those assets, as well as the road network functionality.’ Such information is valuable to enhance the quality of the manuscript. Please consider adding it.

Q4.5 – Then ‘…we can more fully explore how different adaptations may—or may not—contribute to a more equitable and resilient coastal future.’. The sentence creates the expectation of a conclusion, but no further result or detail is given. Which (also) leads to the lack of conclusions in the Conclusion section.

Major Q5 – Conclusions. The Conclusion section does not really provide a conclusion, and half of it addresses ‘everyday disasters’. While the whole manuscript focused on extreme, rare, catastrophic megaquakes and secondary tsunamis, half the remarks propose everyday disasters, such as ‘affordable housing’. Why introduce a new and unrelated topic? The Conclusion section needs substantial leverage, including remarks from the CSZ case-study, discussing future work, etc.

Major Q6 – The adequacy and quality of the figures is insufficient. Fig. 1 shows GEJE simulations, Fig. 2 shows tectonic characteristics of CSZ. Are these (low resolution) figures necessary to the understanding of the framework and conclusions?

Minor comments

Minor Q1 – ‘mitigation strategies for coastal hazards’ – coastal hazards cannot be mitigated, only coastal risk or coastal hazard effects. Please verify hazard and risk concepts, for example on Terminology on disaster risk reduction published by UNISDR.

Minor Q2 – ‘… these compounding disasters…’ – please verify multi-risk concepts and how its inter-dependencies contribute to increasing their potential. Examples:

• Marzocchi W, Garcia-Aristizabal A, Gasparini P, Mastellone ML, Ruocco AD. Basic principles of multi-risk assessment: a case study in Italy. Nat Hazards 2012;62(2):551–73. doi:10.1007/s11069-012-0092-x.

• Selva J. Long-term multi-risk assessment: statistical treatment of interaction among

risks. Nat Hazards 2013;67(2):701–22. doi:10.1007/s11069-013-0599-9.

• Mignan A, Wiemer S, Giardini D. The quantification of low-probability-high-consequences events: Part I. A generic multi-risk approach. Nat Hazards 2014;73(3):1999–2022. doi:10.1007/s11069-014-1178-4.

• Liu Z, Nadim F, Garcia-Aristizabal A, Mignan A, Fleming K, Luna BQ. A three-level framework for multi-risk assessment. Georisk Assess Manage Risk Eng Syst Geohazards 2015;9(2):59–74. doi:10.1080/17499518.2015.1041989.

• Ming X, Xu W, Li Y, Du J, Liu B, Shi P. Quantitative multi-hazard risk assessment with vulnerability surface and hazard joint return period. Stoch Environ Res Risk Assess 2015;29(1):35–44. doi:10.1007/s00477-014-0935-y.

• Reis, C., Lopes, M., Baptista, M. A., & Clain, S. (2022). Towards an integrated framework for the risk assessment of coastal structures exposed to earthquake and tsunami hazards. Resilient Cities and Structures, 1(2), 57-75. doi:10.1016/j.rcns.2022.07.001

Minor Q3 – ‘…tons of disaster debris2. 2At the time of writing this paper the official death toll from Japan’s National Police Agency stood at 15,895, with 2,539 people remaining missing, for a total of 18,434 lives lost in total. The ensuing tsunami was the largest ever recorded in Japan, with a runup reaching 40 meters in some locations (Aldrich, 2019).’ Almost twelve years have passed since GEJE, 2011, which is the most documented natural event in history. There’s no need to use such writing artifacts.

Minor Q4 – If the authors decide to maintain the structure of the manuscript, the definition of the Target Universalism principle would make sense to be part of the Key Concepts section.

Minor Q5 – Missing references. I have spotted Kovanen et al. 2020, but please verify the coherence between citations in the manuscript and list of bib references.

Minor Q6 – In advance, I apologize if my comment should be addressed to the journal rather than the authors, but numbering the lines of the manuscript really makes the reviewer’s job easier.

---

## [Reviewer Report]

Abstract:

The first sentence is very long. Can it be broken into 2 or more sentences for readability?

Delete “To do so we draw upon and just say Drawing on the...and justice, we illustrate...”

Introduction and Background:

Paragraph 1

Replace mount with increase

Can you include newer citations than Adger 2000 and Mimura 2008?

Delete the comma after climate change adaption and before “and acute hazard preparations”

Add a period after the Lipiec citation. Delete particularly and re-write the last sentence.

Paragraph 2

No need to say In order to...just say To meet

Delete henceforth and just include the acronym for GEJE and CSZ

Paragraph 3:

Replace We argue that coastal adaptation with “To do this adaptation policies need to be co-produced...”

Key Concepts:

Equity and Justice: Start a new paragraph with “Each of these subcategories...”

Equitable Resilience: replace existing with “pre-event”

Delete “we, along with a multitude of other scholars and stakeholders argue that”

Operationalizing Equitable Resilience: Start a new paragraph at “Co-production of knowledge operationalizes...”

Delete “In summary”

The Great Japan Earthquake:

Start a new paragraph “At the time of these compounding..”

Add a comma in the same sentence after disasters

I understand that LGBT+ was the focus of this study, but can you add a sentence about other potentially marginalized communities and then edit to say something like - we will now focus on

Add a comma after Because they are less likely to have children, they tend

Co-Production of Natural Hazard Adaptation:

The material was supplemented with secondary materials (not literature as stated)

Add “primary” to “and analyzed alongside local primary data”

The sentence “The study underscored..” should be moved to the discussion / conclusion

Again, delete In order to and just start with To cultivate

town-hall style meetings are not exactly an innovative, power-equitable setting

The sentences that starts with “This co-productive process generated dynamic” should be in discussion or conclusion.

Overall, in this section it is very unclear what of this is new for this publication and what of this is just a summary of the Fox work. If a summary of Fox, it should be described briefly and cited, not repeat extensive sections of prior work.

The paragraph beginning with “Key findings” - are these new? Or again, just need a brief summary of Fox and a citation? All of Page 8 seems to be reporting of Fox’s prior findings.

Delete It is equally significant to note, however and just start with “The ways in which the....”

Applying the Tohoku Model:

Start a new paragraph after daunting.

Add a comma after Bill 379 was repealed in 2019, leaving

The discussion of the % of Latinx residents in certain fields can be shortened and sentences should not start with 32% - could just say “primaily low wage, with 32% of the employees in agriculture...being Latinx”

Same with the sentence that starts with 18%

Delete “it goes without saying that fire, police stations, hospitals, and other” and just say "While critical facilities are essential...'

Delete “To answer this call...” and replace with “We expanded..”

Co-Production with Latinx:

Delete “that identified the importance of community assets”

This is what the paper should be focused on, however, no real results from the Latinx work are provided. The Conclusion says the Cascadia study “illustrates” but really no results were presented.

My overall suggestion is to delete the section on Japan or limit it to a brief review of the prior project. Be clear about the application of the model to Oregon and the framework used - there are important findings - that vulnerable Latinx workers would prefer to prioritize resilience of community organizations over typically critical infrastructure, but that is lost in the long repeat of the Japan data and the lack of a typical “Results” section from the Cascadia study.

---

## [Editor Report]

Dear Dr Fox,

I have now received two detailed reviews of the manuscript you submitted to Coastal Futures. Both reviewers provide detailed comments and suggestions for change. Although the paper makes for interesting reading, I also agree with one of the reviewers that the manuscript in its current form and outline is neither an adequate review of the state of the art nor does it present sufficient analysis to be a research paper.

I am recommending that you consider the detailed comments from the reviewers and re-submit after a major revision.

---

## [Reviewer Report]

Thank you for considering (most) of my suggestions. If other reviewers and/or the editor ask for enhancing Figure 3 quality, I will reiterate their request.

Otherwise, the new manuscript shows an overall improvement that I consider satisfactory. I recommend the manuscript for publication.

---

## [Editor Report]

Dear Dr Fox,

Following the second review by one of the first-round reviewers I am satisfied to recommend that the manuscript be considered for publication following minor revisions. It is however important to note that some changes may require some major rethinking. Overall, the response from the authors was adequate based on the first round of reviews. The revised version was considerably easier to read and also resulted in a few more follow-up questions. I hope that your response will further clarify some issues.

Please note some language editing comments and suggestions in the attached manuscript.

I also provide questions for clarity in the manuscript.

Then, I would like to state again that this work should be published. It will make a valuable contribution to more human-oriented thinking about hazard planning/preparedness. I found the manuscript interesting and it raised many questions with regard to the solutions to the barriers causing inequality in hazard planning. The combination of social challenges with those of the impacts of hazards is interesting but also requires clarity with regard to consistency in terminology.

The objective states the review focuses on “coastal hazard planning”, which is different from “coastal hazard adaptation” in the title. Planning and adaptation may not always mean the same thing.

“Coastal hazard adaptation” (i.e., adaptation, in the title), “coastal hazard planning” (planning of? used in the manuscript), “disaster planning practices”, “disaster mitigation”, “disaster experiences”, “disaster recover” are all used in the manuscript. It was often confusing what element of “hazard” or “disaster” the manuscript focused on, and why. Sometimes it deals with planning, in other instances it discusses responses and recovery. This links with the issue of the temporal scale in the next point.

There is something fundamentally unclear about the definition of “coastal hazard adaptation”, and the way this concept is applied in the manuscript. At the very least, the manuscript should consider the temporal aspect of disasters, from planning to disaster to different stages post-disaster. It seems clear that time and urgency will play a role in post-disaster responses. But then, this can be avoided by planning corrections etc. The temporal scale of the elements that would be included in “coastal hazard adaptation” seems important. Still, the manuscript seamlessly switches between different aspects of “coastal hazard adaptation” i.e., disaster planning, disaster responses, and post-disaster recovery. I can clearly see the need to address social inequality as a systemic issue influencing overall planning, and the same patterns of inequality having a different disaster response (during and immediately after and event), and then again during the recovery phase and access to resources. Surely this understanding and definition of stages of the disaster reduction process are relevant and important?

Both the points above can be dealt with by clear definitions of the objective of the manuscript.

Would the authors have any comment on the limits of fragmentation/differentiation of specific disaster responses for different communities? See the in-text comments.

Targeted universalism is presented as the only framework useful for the context of the paper, and then using one paper (Powell et al 2019). Either acknowledge other potential frameworks and suggest why TU is the most appropriate, or expand the theory of TU to make its selection as a framework more rigorous.

The same argument goes for the selection of Agent-based modelling (ABM) as an “alternative futures support tool”. I would recommend identifying other potential tools before the authors motivate the utility of ABM.

The vignettes are different in scope. One deals with disaster responses and experiences while the other deals with planning. If this was the intent it would be good to acknowledge this upfront.

Figures 1 & 2 can probably be merged into a single figure. Their contribution to the text is limited. Figure 3 can be simplified. It is currently very text-heavy.

The section on “Disaster experiences” not being equal can benefit from a table with additional information. See in-line comment in the manuscript.

Regards

Louis Celliers

---

## [Editor Report]

Dear Dr Fox,

Thank you for your constructive response to the editorial and review comments. I am happy to recommend the publication of the paper. I attach a marked-up version with minor editorial changes.

Regards

Louis Celliers

---

## [Editor Report]

Dear Dr Fox,

I am happy to recommend the publication of your article.

Regards

Louis Celliers